# ECO: Efficient Computational Optimization for Exact Machine Unlearning in Deep Neural Networks

Yu-Ting Huang [1]   Pei-Yuan Wu [2]   Chuan-Ju Wang [3]

## Abstract

This paper introduces ECO, an efficient computational optimization framework that adapts the CP algorithm—originally proposed by Cauwenberghs & Poggio (2000)—for exact unlearning within deep neural network (DNN) models. ECO utilizes a single model architecture that integrates a DNN-based feature transformation function with the CP algorithm, facilitating precise data removal without necessitating full model retraining. We demonstrate that ECO not only boosts efficiency but also maintains the performance of the original base DNN model, and surprisingly, it even surpasses naive retraining in effectiveness. Crucially, we are the first to adapt the CP algorithm's decremental learning for leave-one-out evaluation to achieve exact unlearning in DNN models by fully removing a specific data instance's influence. We plan to open-source our implementation to promote further research in the machine unlearning field.

## 1. Introduction

The concept of *machine unlearning* involves a systematic approach to eliminate the influence of specific data points from a previously trained machine learning model. This procedure has recently been legally mandated by several regulations, including the California Consumer Privacy Act (CCPA) (de la Torre, 2018), Canada's Consumer Privacy Protection Act (CPPA), the European Union's General Data Protection Regulation (GDPR) and the former Right to Be Forgotten (Chris Jay Hoofnagle & Borgesius, 2019; Rosen, 2012). These laws mandate that companies and organizations enable users to withdraw consent for their data at any time, which requires the removal of this data's influence from any machine learning models used. To comply with these requirements, one common strategy is *naive retraining*, which entails retraining models by utilizing the remaining training data after excluding the data points that have been requested for removal. However, this solution frequently incurs substantial computation and time overhead.

To address the limitations of naive retraining, various strategies for machine unlearning have been developed, showing promising results in efficiently removing data influence. Previous research in machine unlearning has explored algorithms not based on deep neural networks (DNNs). Notable examples include unlearning methods for linear models (Mahadevan & Mathioudakis, 2021), support vector machine (SVM) (Cauwenberghs & Poggio, 2000), K-means (Ginart et al., 2019), and random forest (Brophy & Lowd, 2021).

For DNN-based machine learning algorithms, research into machine unlearning has branched into two main approaches: *exact unlearning* and *approximate unlearning*. Several approximate unlearning methods utilize the Fisher information matrix (FIM) and influence functions to facilitate data removal (Guo et al., 2020; Sekhari et al., 2021). Alternative approaches include variational forgetting for regression and Gaussian processes (Nguyen et al., 2020), neural tangent kernel forgetting (NTK) (Golatkar et al., 2020), and mixed-linear models (MLM) (Golatkar et al., 2021).

Despite the purported computational benefits of these approximation methods,[1] they often lack a robust guarantee of the unlearning process's efficacy, leading to a performance gap compared to exact unlearning (Jia et al., 2023). Moreover, these methods have not consistently demonstrated reliability in terms of forgetting quality. In contrast, exact unlearning methods inherently offer reliability by directly eliminating the requested data. Research in this area aims to optimize the retraining process to reduce costs (Bourtoule et al., 2021; Yan et al., 2022). For example, SISA (Bourtoule et al., 2021) utilizes data partitioning mechanisms

---

[1]Graduate Program of Data Science, National Taiwan University and Academia Sinica, Taipei, Taiwan [2]National Taiwan University, Taipei, Taiwan [3]Academia Sinica, Taipei, Taiwan. Correspondence to: Yu-Ting Huang <r11946008@ntu.edu.tw>, Chuan-Ju Wang <cjwang@citi.sinica.edu.tw>.

Accepted to the Workshop on Advancing Neural Network Training at International Conference on Machine Learning (WANT@ICML 2024).

---

[1]Some recent studies, including (Jia et al., 2023; Foster et al., 2024; Zhang et al., 2022), have shown that the computation time for these approximate unlearning methods may be even longer than initially anticipated.

to accelerate retraining, segmenting datasets into distinct, non-overlapping shards and training a set of weak learners for each shard. However, such research typically entails labor-intensive tasks, including data partitioning, ensembling multiple sub-models, and tracking the order of data training—complexities that challenge service providers in maintaining an efficient workflow. Moreover, these exact unlearning methods generally yield lower model accuracy compared to naive retraining (Bourtoule et al., 2021; Yan et al., 2022; Shen et al., 2024; He et al., 2021; Zhang et al., 2022; Li et al., 2023; Yan et al., 2022).

In supervised learning, the addition or removal of small data amounts typically results in minimal changes to a model's core characteristics. This need has led to the development of incremental and decremental algorithms that update models efficiently without full retraining. Research in the dual problem of kernel SVMs, including the CP algorithm (Cauwenberghs & Poggio, 2000) and subsequent studies (Laskov et al., 2006; Karasuyama & Takeuchi, 2009), has explored methods to maintain optimality when modifying the training set. However, due to the understanding and implementation challenges of such algorithms discussed in (Laskov et al., 2006) and the complexities of DNN models, applying these methods directly to DNN models is not trivial, marking an important area for further exploration.

To this end, we develop an efficient computational optimization framework (ECO) that adapts the CP algorithm for exact unlearning in the context of DNN models. Unlike previous efforts that rely on multiple weak learners (Bourtoule et al., 2021; Yan et al., 2022), our approach uses a single model for effective unlearning. Our key innovations include: 1) employing the CP algorithm to identify a significantly smaller *core dataset* compared to the original training set; 2) enhancing the traditional CP algorithm, which typically uses predefined kernels, by integrating DNNs to create a data-driven feature transformation function. Our method is a hybrid approach that employs two distinct optimizers: the CP algorithm for the final fully connected layer and a standard gradient descent optimizer for the DNN-based feature transformation function. This dual optimization approach allows for exact unlearning without retraining the entire model, promoting both optimality and efficiency. The contributions of this paper can be summarized as follows:

- We present ECO, an innovative hybrid framework that integrates the CP algorithm with DNNs for exact machine unlearning. This approach simplifies labor-intensive tasks, significantly reducing the workload for service providers compared to previous exact unlearning methods for DNNs.

- Our experiments show that the proposed hybrid approach, ECO, not only reliably removes the requested data but also maintains, and surprisingly enhances, the base DNN model's performance compared to methods using only a gradient descent-based optimizer.

- We are the first to adapt the CP algorithm's leave-one-out decremental learning for exact unlearning in DNNs, making a significant advancement in the field. We also open-source a usable base code for the CP algorithm, addressing the previous lack of such resources and encouraging further research and practical applications.

## 2. Problem Definition and Preliminaries

### 2.1. Problem Definition

Machine unlearning aims to eliminate the influence of specific training data from an already-trained machine learning model (Cao & Yang, 2015; Bourtoule et al., 2021). Let $\mathcal{D}_{\text{train}}$ be a traning set, and $\mathcal{D}_f \subseteq \mathcal{D}_{\text{train}}$ the *forget set*, which includes the instances targeted for removal. The remaining data, known as the *retained set*, and denoted as $\mathcal{D}_r = \mathcal{D}_{\text{train}} \setminus \mathcal{D}_f$, are those instances whose influence is to be preserved in the model. We denote the model post-unlearning as $\theta_u$, which is adjusted to exclude the impacts of $\mathcal{D}_f$ from the original model $\theta_o$, trained on the full dataset $\mathcal{D}_{\text{train}}$. The primary challenge of machine unlearning is to develop an efficient and effective method to transition from $\theta_o$ to $\theta_u$. In our analysis, we assess the efficiency of the unlearning process by measuring the time required for its completion. Additionally, we evaluate the efficacy of the unlearned model, $\theta_u$, by examining its performance with respect to the retained set, $\mathcal{D}_r$, the forget set, $\mathcal{D}_f$, and an independent *test set*, $\mathcal{D}_{\text{test}}$, which does not overlap with the training set $\mathcal{D}_{\text{train}}$.

### 2.2. Revisit the Dual SVM and the CP Algorithm

Given a training dataset $\{(x_i, y_i)\}_{i=1}^n$, where $x_i \in \mathcal{X} \subseteq \mathbb{R}^d$ represents the input features and $y_i \in \{-1, +1\}$ denotes the output class label, the dual SVM learns the prediction function $f(\cdot)$ using learnable parameter $\alpha_i$ for each data point $x_i$ and a bias term $b$. The prediction function is defined as:

$$f(x) = \sum_{i=1}^n \alpha_i y_i \langle \Phi(x_i), \Phi(x) \rangle + b, \qquad (1)$$

where $\Phi(.)$ is a function mapping $x$ into a latent space, and $\langle \cdot, \cdot \rangle$ denotes the inner product in this space. Based on the Karush-Kuhn-Tucker (KKT) optimality conditions, the optimization process for the dual SVM must satisfy the following two criteria. First,

$$\sum_{i=1}^n y_i \alpha_i = 0, \qquad (2)$$

and, secondly, all data points are classified into the three groups according to their classification margins and corre-

sponding Lagrange multipliers $\alpha_i$ (Cauwenberghs & Poggio, 2000; Laskov et al., 2006; Karasuyama & Takeuchi, 2009):

$$\mathcal{O} = \{i|y_i f(x_i) > 1, \alpha_i = 0\}, \tag{3}$$

$$\mathcal{M} = \{i|y_i f(x_i) = 1, \alpha_i \in [0, C]\}, \tag{4}$$

$$\mathcal{I} = \{i|y_i f(x_i) < 1, \alpha_i = C\}, \tag{5}$$

where $C \in \mathbb{R}^+$ is the predefined regularization parameter. From (1), it is clear that assigning a value of 0 to $\alpha_i$ effectively removes the data point $(x_i, y_i)$ from the prediction function. In the context of machine unlearning, this action serves to unlearn $(x_i, y_i)$ from the model. Notably, this method of unlearning is *exact*, eliminating any influence of $(x_i, y_i)$ on the model.

Following this concept, we provide a concise overview of how the algorithm proposed by Cauwenberghs & Poggio (2000)—hereafter referred to as the CP algorithm—updates an exact optimal solution to maintain the KKT optimality conditions. This update is performed without the need for model retraining when learning or unlearning specific examples. We denote the difference between the updated value of a variable $h$ and its original value as $\Delta h$ and refer to the data currently being learned or unlearned as $(x_c, y_c)$. Given that $\alpha_i$ values for instances in $\mathcal{M}$ (as described in (4)) can vary from 0 to $C$, they exhibit a high degree of flexibility. Therefore, the strategy involves adjusting $\alpha_i$ of instances in $\mathcal{M} = \{m_1, m_2, \cdots, m_{|\mathcal{M}|}\}$ to accommodate $\Delta \alpha_c$ while still upholding the KKT condition. By incorporating these adjustments into (2) and (4), we derive the following equation to maintain KKT compliance:

$$\underbrace{\begin{bmatrix} 0 & y_{m_1} & \cdots & y_{m_{|\mathcal{M}|}} \\ y_{m_1} & d_{m_1 m_1} & \cdots & d_{m_1 m_{|\mathcal{M}|}} \\ \vdots & \vdots & \ddots & \vdots \\ y_{m_{|\mathcal{M}|}} & d_{m_{|\mathcal{M}|} m_1} & \cdots & d_{m_{|\mathcal{M}|} m_{|\mathcal{M}|}} \end{bmatrix}}_{Q} \begin{bmatrix} \Delta b \\ \Delta \alpha_{m_1} \\ \vdots \\ \Delta \alpha_{m_{|\mathcal{M}|}} \end{bmatrix}$$
$$= - \begin{bmatrix} y_c \\ d_{m_1 c} \\ \vdots \\ d_{m_{|\mathcal{M}|} c} \end{bmatrix} \Delta \alpha_c, \tag{6}$$

where $d_{ij} = y_i y_j \langle \Phi(x_i), \Phi(x_j) \rangle$. The goal of the CP algorithm is to find the appropriate $\alpha_i$ value assignments that satisfy the KKT condition for datasets that are enlarged $(n \leftarrow n+1)$ or reduced $(n \leftarrow n-1)$. More details of the CP algorithms can be found in Cauwenberghs & Poggio (2000) and in 3.2.1.

# 3. Our Proposed Method: ECO

Unlike previous exact unlearning efforts that relied on numerous weak learners (Bourtoule et al., 2021; Yan et al., 2022), our approach, ECO, utilizes a single model to effi-

ciently and effectively achieve unlearning. The key innovations of ECO include: 1) utilizing the CP algorithm to identify a significantly smaller *core dataset*, denoted as $\mathcal{C}_{\mathrm{CP}}$, compared to the original training set $\mathcal{D}_{\mathrm{train}}$; 2) enhancing the traditional CP algorithm, which typically employs predefined hand-crafted kernels, by incorporating deep neural networks (DNNs) to develop a data-driven feature transformation function $\Phi(.)$, trained exclusively on the core dataset.

ECO employs two distinct optimizers: the CP algorithm for the final fully connected layer and a classic gradient-descent-based optimizer for the feature transformation function $\Phi(x)$. This dual optimization strategy enables *exact* unlearning without retraining the entire model, thus ensuring both optimality and efficiency. Furthermore, since $\Phi(.)$ is trained exclusively on the core dataset, updates are only necessary if the forget set $\mathcal{D}_f$ intersects with it. As the core dataset is significantly smaller than the original $\mathcal{D}_{\mathrm{train}}$, such intersections are less likely, further enhancing the efficiency of our proposed method.

Below, we outline our method in two main parts: 1) model preparation and 2) model serving. In the subsequent sections, we detail each component and demonstrate how our proposed framework overcomes challenges encountered in earlier methods. Note that while we use binary classification to illustrate our method, this methodology can be straightforwardly extended to multi-class classification and regression tasks.

## 3.1. Model Preparation

Consider a DNN model, trained on the training set $\mathcal{D}_{\mathrm{train}}$, defined by:

$$f_{\mathcal{D}_{\mathrm{train}}}(x; \tilde{W}_{\mathcal{D}_{\mathrm{train}}}, \tilde{b}_{\mathcal{D}_{\mathrm{train}}}, \Phi_{\mathcal{D}_{\mathrm{train}}})$$
$$= \tilde{W}_{\mathcal{D}_{\mathrm{train}}}^{\mathsf{T}} \Phi_{\mathcal{D}_{\mathrm{train}}}(x) + \tilde{b}_{\mathcal{D}_{\mathrm{train}}}, \tag{7}$$

using a gradient-based (GD-based) optimizer and a cross-entropy loss function $\ell_{\mathrm{CNT}}$. Based on the form outlined in (1), our initial step utilizes the CP algorithm to find a functionally equivalent alternative to $f_{\mathcal{D}_{\mathrm{train}}}$, expressed as

$$\tilde{f}_{\mathcal{D}_{\mathrm{train}}}(x; \alpha, b, \Phi_{\mathcal{D}_{\mathrm{train}}})$$
$$= \sum_{i=1}^{n} \alpha_i y_i \Phi_{\mathcal{D}_{\mathrm{train}}}(x_i)^{\mathsf{T}} \Phi_{\mathcal{D}_{\mathrm{train}}}(x) + b. \tag{8}$$

Above, (8) replaces $\tilde{W}_{\mathcal{D}_{\mathrm{train}}}$ and $\tilde{b}_{\mathcal{D}_{\mathrm{train}}}$ in (7) with $\sum_{i=1}^{n} \alpha_i y_i \Phi_{\mathcal{D}_{\mathrm{train}}}(x_i)$ and $b$, respectively. Note that in this step, we freeze $\Phi_{\mathcal{D}_{\mathrm{train}}}(\cdot)$ and employ the CP learning algorithm (refer to Algorithm 1 in Cauwenberghs & Poggio (2000)) to learn $\alpha_i$ for each instance $x_i \in \mathcal{D}_{\mathrm{train}}$, as well as the bias term $b$. Recall from (8) that our method utilizes two distinct optimizers: the CP algorithm for the final fully con-

nected layer and a classic gradient-descent-based optimizer for the feature transformation function $\Phi(x)$.

Using (8), we divide the training instances in $\mathcal{D}_{\text{train}}$ into three groups: $\mathcal{O}, \mathcal{M}$, and $\mathcal{I}$ based on (3)–(5). We then proceed to identify the core dataset $\mathcal{C}_{\text{CP}}$ as:

$$\mathcal{C}_{\text{CP}} = \mathcal{M} \cup \mathcal{I} \cup \tilde{\mathcal{O}}, \tag{9}$$

where $\tilde{\mathcal{O}}$ denotes the set of instances with the lowest $k_{\text{CP}}$ values of $y_i f(x_i)$ for $i \in \mathcal{O}$. The rationale behind this configuration is to focus on instances that are not well-learned. Specifically, from (3) to (5), it is clear that $\mathcal{M}$ and $\mathcal{I}$ encompass data points where $y_i f(x_i) \leq 1$, signaling suboptimal prediction. Moreover, we refine our selection by including instances from $\mathcal{O}$ in the construction of $\tilde{\mathcal{O}}$ that exhibit lower values of $y_i f(x_i)$, denoting relatively poorer quality predictions.

In the final stage of model preparation, we aim to ensure that the prediction function depends exclusively on the core dataset, $\mathcal{C}_{\text{CP}}$. To this end, we pose the following minimization problem:

$$\mathbb{E}_{x \sim \mathcal{C}_{\text{CP}}} \left[ \ell_{\text{MSE}}(\Phi_{\mathcal{C}_{\text{CP}}}(x), \Phi_{\mathcal{D}_{\text{train}}}(x)) \right] \tag{10}$$

to develop a new transformation function $\Phi_{\mathcal{C}_{\text{CP}}}(x)$.

Above, we employ data points in $\mathcal{C}_{\text{CP}}$ to train a $\Phi_{\mathcal{C}_{\text{CP}}}(x)$ that mimics $\Phi_{\mathcal{D}_{\text{train}}}(x)$ using a mean squared error (MSE) loss function, $\ell_{\text{MSE}}$, and a classic gradient-descent-based optimizer, resulting in

$$f_{\mathcal{C}_{\text{CP}}}(x; \alpha, b, \Phi_{\mathcal{C}_{\text{CP}}}) = \sum_{i \in \mathcal{C}_{\text{CP}}} \alpha_i y_i \Phi_{\mathcal{C}_{\text{CP}}}(x_i)^\intercal \Phi_{\mathcal{C}_{\text{CP}}}(x) + b. \tag{11}$$

Note that $\alpha_i > 0$ applies only to instances belonging to $\mathcal{M}$ and $\mathcal{I}$. Therefore, with the newly learned $\Phi_{\mathcal{C}_{\text{CP}}}(x)$, (11) becomes irrelevant to instances outside the set $\mathcal{C}_{\text{CP}}$. The complete procedure of model preparation is outlined in Algorithm 1.

---

**Algorithm 1** ECO: Model Preparation

1: **Input:** $\mathcal{D}_{\text{train}}$
2: **Output:** $f_{\mathcal{C}_{\text{CP}}}(x), \mathcal{C}_{\text{CP}}$
3: $f_{\mathcal{D}_{\text{train}}} \leftarrow$ Use the classic GD-based optimizer for model training with $\mathcal{D}_{\text{train}}$ (see (7))
4: $\alpha, b \leftarrow$ Employ Algorithm 1 in Cauwenberghs & Poggio (2000) with fixed $\Phi_{\mathcal{D}_{\text{train}}}(x)$ to obtain $\alpha_i$ and $b$ in (8)
5: $\mathcal{C}_{\text{CP}} \leftarrow$ Construct the core dataset via (9)
6: $\Phi_{\mathcal{C}_{\text{CP}}} \leftarrow$ Learn a new feature transformation function $\Phi_{\mathcal{C}_{\text{CP}}}(x)$ with the loss in (10)

---

### 3.1.1. A STRATEGY FOR CP LEARNING ACCELERATION

Recall that the initial step in the above model preparation involves using the CP algorithm to identify a functionally

equivalent alternative $\tilde{f}_{\mathcal{D}_{\text{train}}}$ to $f_{\mathcal{D}_{\text{train}}}$, as shown in (8). However, this step requires calculations involving all instances in $\mathcal{D}_{\text{train}}$, leading to inefficiencies. To address this, we propose an approximation strategy that accelerates the optimization process of the CP algorithm.

Specifically, this approximation is implemented before using the CP algorithm to derive (8). The main strategy involves identifying a smaller subset, $\mathcal{C}_{\text{GD}}$, from $\mathcal{D}_{\text{train}}$. This subset consists of instances with the highest $k_{\text{GD}}$ values of losses $\ell_{\text{CNT}}$ from the given DNN model. The rationale behind this design is twofold: 1) first, the prediction function in (1) is solely influenced by data with non-zero $\alpha_i$; 2) the categorization in (3)–(5) indicates that data points with non-zero $\alpha_i$ are those where $y_i f(x_i) \leq 1$, suggesting suboptimal prediction. Given these observations and within the context of DNN models, higher values of $\ell_{\text{CNT}}$ indicate relatively poorer predictions, thereby likely resulting in non-zero $\alpha_i$. Consequently, instead of assigning $\alpha_i$ for the full $\mathcal{D}_{\text{train}}$ using the CP algorithm, we streamline the computational process by focusing solely on this critical yet smaller subset, $\mathcal{C}_{\text{GD}}$ since for instances in $\mathcal{D}_{\text{train}} \setminus \mathcal{C}_{\text{GD}}$, $\alpha_i$ values are most likely zero, making this approach both efficient and effective.

Using $\mathcal{C}_{\text{GD}}$, we train a DNN model $f_{\mathcal{C}_{\text{GD}}}(x)$ by employing the same approach as outlined in (7), using a gradient-based optimizer and a cross-entropy loss function $\ell_{\text{CNT}}$:

$$f_{\mathcal{C}_{\text{GD}}}(x; \tilde{W}_{\mathcal{C}_{\text{GD}}}, \tilde{b}_{\mathcal{C}_{\text{GD}}}, \Phi_{\mathcal{C}_{\text{GD}}}) = \tilde{W}_{\mathcal{C}_{\text{GD}}}^\intercal \Phi_{\mathcal{C}_{\text{GD}}}(x) + \tilde{b}_{\mathcal{C}_{\text{GD}}}. \tag{12}$$

Adopting the methodology from (7) to (8) and substituting $\mathcal{D}_{\text{train}}$ with $\mathcal{C}_{\text{GD}}$, we derive:

$$\tilde{f}_{\mathcal{C}_{\text{GD}}}(x; \alpha, b, \Phi_{\mathcal{C}_{\text{GD}}}) = \sum_{i \in \mathcal{C}_{\text{GD}}} \alpha_i y_i \Phi_{\mathcal{C}_{\text{GD}}}(x_i)^\intercal \Phi_{\mathcal{C}_{\text{GD}}}(x) + b. \tag{13}$$

With (13), the remainder of the model preparation process follows the steps outlined in Lines 5 and 6 of Alorithm 1. Note that different from the original version, the derived core dataset, $\mathcal{C}_{\text{CP}}$, is based not on the categorization of $\mathcal{D}_{\text{train}}$ but on the subset $\mathcal{C}_{\text{GD}}$.

### 3.2. Model Serving

We now demonstrate the process for handling an unlearning request, $\mathcal{D}_f \subset \mathcal{D}_{\text{train}}$. Since the model $f_{\mathcal{C}_{\text{CP}}}(x)$, detailed in (11), is solely derived from $\mathcal{C}_{\text{CP}}$, updates are necessary only if the intersection $\mathcal{D}_f \cap \mathcal{C}_{\text{CP}}$ is non-empty. When a request to unlearn $\mathcal{D}_f$ arises, the model update procedure is segmented into two main steps: 1) the last fully connected layer and 2) all other layers except the last fully connected layer, i.e., the feature transformation function $\Phi_{\mathcal{C}_{\text{CP}}}(x)$. First, for the last fully connected layer, which is influenced solely by $\mathcal{M}$ and $\mathcal{I}$ (as indicated from (3) to (5)) where $\alpha_i > 0$, we update it only if $\mathcal{D}_f$ intersects with $\mathcal{M} \cup \mathcal{I}$. This update is performed

using a modified version of the original CP unlearning algorithm outlined in Section 3.2.1.[2] Subsequently, we revise $\mathcal{C}_{\text{CP}}$ via (9) based on the newly identified $\mathcal{M}$, $\mathcal{I}$, and $\tilde{\mathcal{O}}$.

After the first step, for $\Phi_{\mathcal{C}_{\text{CP}}}(x)$, updates are necessary only if $\mathcal{D}_f \cap \mathcal{C}_{\text{CP}} \neq \emptyset$ and performed through the optimization in (10). Since the size of $\mathcal{C}_{\text{CP}}$ is considerably smaller than that of $\mathcal{D}_{\text{train}}$, the need for unlearning is reduced, and often the model remains unaffected by $\mathcal{D}_f$.

Note that the above two steps must be performed sequentially because the set $\mathcal{C}_{\text{CP}}$ is determined by the CP algorithm; reversing the order would result in logical inconsistencies. Algorithm 2 illustrates how our model processes an unlearning request, resulting in an unlearned model $\theta_u \equiv f_{\mathcal{C}_{\text{CP}}}(x)$.

---

**Algorithm 2** ECO: Model Serving

1: **Input:** $\mathcal{D}_f$, $f_{\mathcal{C}_{\text{CP}}}(x)$, $\mathcal{C}_{\text{CP}}$
2: **Output:** $f_{\mathcal{C}_{\text{CP}}}(x)$, $\mathcal{C}_{\text{CP}}$
3: **if** $\mathcal{D}_f \cap \mathcal{C}_{\text{CP}} \neq \emptyset$ **then**
4:     **if** $\mathcal{D}_f \cap (\mathcal{M} \cup \mathcal{I}) \neq \emptyset$ **then**
5:         $\alpha, b \leftarrow$ Employ Algorithm 3 to unlearn $\mathcal{D}_f$
6:         $\mathcal{C}_{\text{CP}} \leftarrow$ Construct the new core dataset via (9)
7:     **else**
8:         $\mathcal{C}_{\text{CP}} \leftarrow \mathcal{C}_{\text{CP}} \setminus \mathcal{D}_f$
9:     **end if**
10:    $\Phi_{\mathcal{C}_{\text{CP}}} \leftarrow$ Learn a new feature transformation function $\Phi_{\mathcal{C}_{\text{CP}}}(x)$ with the loss in (10)
11: **else**
12:    Remain the input model $f_{\mathcal{C}_{\text{CP}}}(x)$ and the set $\mathcal{C}_{\text{CP}}$
13: **end if**

---

### 3.2.1. THE UNLEARNING PROCEDURE OF THE CP ALGORITHM

Originally, the decremental learning algorithm from Cauwenberghs & Poggio (2000) was designed to accelerate leave-one-out cross-validation (Brophy & Lowd, 2021; Laskov et al., 2006; Chen et al., 2022; Nguyen et al., 2022; Guo et al., 2020). Moreover, Algorithm 2 in Cauwenberghs & Poggio (2000) does not fully address the needs of exact machine unlearning. To this end, we modify this original algorithm to ensure completeness in machine unlearning, specifically by adjusting the $\alpha_i$ values of unlearned data instances to zero. Before presenting the modified algorithm, we first outline the updating rules of the CP algorithm as follows.

Recall that the strategy of the CP algorithm involves adjusting $\alpha_i$ of instances in $\mathcal{M}$ to accommodate $\Delta\alpha_c$ while still upholding the KKT condition, resulting in (6) in Section 2.2. Under the assumption that matrix $Q$ is invertible (Cauwen-

---

[2]The original decremental learning algorithm of the CP algorithm (see Algorithm 2 in Cauwenberghs & Poggio (2000)) was initially designed for leave-one-out validation.

---

berghs & Poggio, 2000), we establish the update rule for $b$ and all $\alpha_j$ of $\mathcal{M}$ as follows:

$$\begin{bmatrix} \Delta b \\ \Delta\alpha_{m_1} \\ \vdots \\ \Delta\alpha_{m_{|\mathcal{M}|}} \end{bmatrix} = \beta\Delta\alpha_c, \tag{14}$$

where

$$\beta = \begin{bmatrix} \beta_0 \\ \beta_{m_1} \\ \vdots \\ \beta_{m_{|\mathcal{M}|}} \end{bmatrix} = -Q^{-1} \begin{bmatrix} y_c \\ d_{m_1 c} \\ \vdots \\ d_{m_{|\mathcal{M}|} c} \end{bmatrix},$$

and each $d_{m_i c}$ represents the interaction between the current instance and the instances in $\mathcal{M}$.

---

**Algorithm 3** CP Unlearning ($n \rightarrow n-1$)

1: **if** $\alpha_c = 0$ **then**
2:    Do nothing.
3: **else**
4:    **while** $\alpha_c > 0$ **do**
5:       $\Delta\alpha_c \leftarrow$ Calculate the largest possible decrement $\alpha_c$ so that elements of training dataset $\{(x_i, y_i)\}_{i=1}^n$ migrate across $\mathcal{I}$, $\mathcal{M}$, and $\mathcal{O}$
6:       $\alpha_c \leftarrow \alpha_c + \Delta\alpha_c$
7:       $\alpha_j, b \leftarrow$ Employ (14) to obtain $\alpha_j, b$
8:       $g_i \leftarrow$ Employ (16) to obtain $g_i$
9:    **end while**
10: **end if**

---

Following Cauwenberghs & Poggio (2000), we define $g_i$ as the first-order derivative of the convex quadratic objective function in the dual SVM with respect to $\alpha_i$; with the definition of $d_{ij}$ (see (6)), we have

$$g_i = y_i f(x_i) - 1 = \sum_{j=1}^n d_{ij}\alpha_j + y_i b - 1.$$

Furthermore, with data $(x_c, y_c)$ being unlearned, $g_i$ updates based on $\Delta\alpha_c$, $\Delta\alpha_j$ and $\Delta b$:

$$\Delta g_i = d_{ic}\Delta\alpha_c + \sum_{j \in \mathcal{M}} d_{ij}\Delta\alpha_j + y_i\Delta b. \tag{15}$$

By substituting (14) into (15), we derive the following update rule for $g_i$:

$$\Delta g_i = d_{ic}\Delta\alpha_c + \sum_{j \in \mathcal{M}} d_{ij}\beta_j\Delta\alpha_c + y_i\beta_0\Delta\alpha_c$$
$$= \Delta\alpha_c \underbrace{\left( d_{ic} + \sum_{j \in \mathcal{M}} d_{ij}\beta_j + y_i\beta_0 \right)}_{\gamma_i}. \tag{16}$$

Note that $\Delta\alpha_c$ is the sole driver influencing the update rules in both (14) and (16). With (14), we monitor if data instances in $\mathcal{M}$ move to $\mathcal{I}$ or $\mathcal{O}$. Conversely, with (16), we monitor data instances if data instances in $\mathcal{I}$ or $\mathcal{O}$ move into $\mathcal{M}$. With the above notations, we provide a high-level overview of the modified CP unlearning process in Algorithm 3. Note that in this unlearning process, updates are necessary for $\mathcal{O}$, $\mathcal{M}$, $\mathcal{I}$, $Q$, $Q^{-1}$, $\beta$ and $\gamma_i$.

# 4. Experiments

## 4.1. Dataset and Models Used

We conduct experiments using LeNet5 on the MNIST handwritten image dataset (Lecun et al., 1998)[3]. It is important to highlight that, despite the lack of standard benchmarks in the field of machine unlearning, the MNIST dataset is the most frequently used dataset according to statistics in (Nguyen et al., 2022).

## 4.2. Evaluation Metrics

There is consensus that an effective unlearning algorithm should ensure data is forgotten while optimizing time efficiency and maintaining model utility. However, assessing the effectiveness of machine unlearning algorithms involves diverse metrics, currently without a unified standard (Thudi et al., 2022; Kurmanji et al., 2023). In this paper, we adopt the contemporary methodology advocated by Jia et al. (2023); Fan et al. (2024), which promotes a *full-stack* evaluation of machine unlearning, incorporating accuracy, time cost, and forget quality. Specifically, we assess the effectiveness of an unlearning algorithm by evaluating the accuracy of the unlearned model $\theta_u$ (see the definition in Section 2.1) on the retain set $\mathcal{D}_r$, the forget set $\mathcal{D}_f$, and the test set $\mathcal{D}_{\text{test}}$, denoted as $\text{Acc}_r$, $\text{Acc}_f$, and $\text{Acc}_{\text{test}}$, respectively. We also calculate $\text{Acc}_{\text{all}} = \text{Acc}_r \times \text{Acc}_f \times \text{Acc}_{\text{test}}$ for an overall comparison, and we include a time cost (seconds) comparison. Although our method qualifies as exact unlearning, ensuring complete removal of the requested data, we still provide the membership inference attack (MIA) score in (Jia et al., 2023) for reference. All the metrics mentioned above are expressed as $a \pm b$, where '$a$' represents the mean and '$b$' denotes the standard deviation across 10 independent trials with different random seeds.[4] The symbol '↑' indicates that higher values are better, and '↓' indicates that lower values are preferable. The best result is highlighted in bold, and the second-best result is underlined.

---

[3]We use the implementation and adopt the hyperparameters from `https://github.com/rasbt/deeplearning-models/blob/master/pytorch_ipynb/cnn/cnn-lenet5-mnist.ipynb`.

[4]We used 10 random seeds, ranging from 2015 to 2024, for all experiments.

## 4.3. Implementation Details

For the selection of $\mathcal{C}_{\text{GD}}$, as detailed in Section 3.1.1, we select instances with the highest $k_{\text{GD}} = 12000$ values of losses $\ell_{\text{CNT}}$ from the given DNN model. The effects of using different $k_{\text{GD}}$ values are discussed in Section 4.5.3. For constructing $\mathcal{O}$ within $\mathcal{C}_{\text{CP}}$, as specified in (9), we select instances with the lowest $k_{\text{CP}}$ values of $y_i f(x_i)$ for $i \in \mathcal{O}$. This was done so that the size of $\mathcal{C}_{\text{CP}}$ equals $10 \times |\mathcal{M} \cup \mathcal{I}|$, resulting in $k_{\text{CP}} \approx 10000$ in our experiments.[5] Unlike many studies in computer vision, we do not employ any augmentations for model training. Additionally, we halt training if $\text{Acc}_{\text{test}}$ fails to improve for five consecutive evaluations.[6]

## 4.4. Compared Methods

We compare the proposed method, ECO, with two other approaches. The first is naive retaining (denoted as NR), which involves retraining the DNN models upon receiving an unlearning request. Additionally, we compare ECO with the intermediate product of our ECO algorithm, designated as $\text{ECO}_i$, which is the DNN model train on $\mathcal{C}_{\text{GD}}$, as presented in (12). The rationale for including $\text{ECO}_i$ for comparison is to validate the reasonableness of our acceleration methodology introduced in Section 3.1.1 by shrinking the dataset from $\mathcal{D}_{\text{train}}$ to $\mathcal{C}_{\text{GD}}$ through the selection of the highest $k_{\text{GD}}$ loss values of the original DNN model. Note that we do not include the well-known exact unlearning approach, SISA (Bourtoule et al., 2021), in our comparisons as it has reported sacrifices performance compared to NR in many previous studies (Bourtoule et al., 2021; Yan et al., 2022; Shen et al., 2024; He et al., 2021; Zhang et al., 2022; Li et al., 2023; Yan et al., 2022).

## 4.5. Experimental Results

### 4.5.1. Main Results

**In-time Unlearning** We start by exploring the scenario of in-time random sample unlearning as shown in Table 1. In this scenario, each unlearning request must be executed immediately upon receipt from the data owner to prevent potential costs, such as fines or loss of trust in the company. This operational context requires that each sample be unlearned individually as requests are received instead of accumulating them for batch unlearning. Therefore, in

---

[5]In our experiments, as we implement acceleration during the model preparation stage (see Section 3.1.1), it is important to note that the three sets $\mathcal{M}$, $\mathcal{I}$, and $\mathcal{O}$ are categorized from $\mathcal{C}_{\text{GD}}$, not from $\mathcal{D}_{\text{train}}$.

[6]We conduct evaluations every 50 iterations, adhering to the training details provided in the implementation available at `https://github.com/rasbt/deeplearning-models/blob/master/pytorch_ipynb/cnn/cnn-lenet5-mnist.ipynb`.

*Table 1.* In-time unlearning

|  | NR | $ECO_i$ | ECO |
|---|---|---|---|
| $Acc_r \uparrow$ | $0.988 \pm 0.005$ | $\underline{0.996 \pm 0.001}$ | $\mathbf{0.997 \pm 0.001}$ |
| $Acc_f \uparrow$ | $\mathbf{1.0 \pm 0.0}$ | $\mathbf{1.0 \pm 0.0}$ | $\mathbf{1.0 \pm 0.0}$ |
| $Acc_{\text{test}} \uparrow$ | $0.986 \pm 0.002$ | $\mathbf{0.989 \pm 0.001}$ | $\mathbf{0.989 \pm 0.001}$ |
| $Acc_{\text{all}} \uparrow$ | $0.974 \pm 0.007$ | $\underline{0.986 \pm 0.002}$ | $\mathbf{0.987 \pm 0.002}$ |
| Time cost (sec) $\downarrow$ | $20.968 \pm 7.992$ | $\mathbf{1.493 \pm 4.476}$ | $\underline{1.842 \pm 5.524}$ |

each trial using a random seed, we randomly select one instance as $\mathcal{D}_f$ for all compared methods. From the table, it is evident that the proposed ECO achieves the highest model accuracy among the three methods evaluated. Also, the standard deviations for ECO and $ECO_i$ are significantly smaller than those of NR, highlighting the enhanced stability of our framework. Both ECO and its intermediate product, $ECO_i$, substantially decrease the time required to execute unlearning requests compared to NR.

**Off-time Batch Unlearning** In scenarios where immediate unlearning is not imperative, the focus can shift from minimizing time costs to prioritizing the minimization of performance degradation after data removal. To demonstrate this, we conduct simulations involving the accumulation of $p\%$ requested unlearned data, termed "off-time batch unlearning," where $p = \{1, 5, 10, 20, 30, 40, 50\}$. Specifically, in each trial using a random seed, we randomly select $p\% \times |\mathcal{D}_{\text{train}}|$ instances as $\mathcal{D}_f$ for all compared methods. This approach allows for a comprehensive evaluation of performance retention abilities in the context of off-time batch unlearning. As demonstrated in Table 2, our approach ECO consistently surpasses the gold standard NR in accuracy across the forget set and the test set ($Acc_f$ and $Acc_{\text{test}}$, respectively) and shows even superior performance in terms of overall accuracy $Acc_{\text{all}}$. Unexpectedly, $Acc_f$ of our approach significantly exceeds the gold standard by a considerable margin, which is remarkable. These results suggest a reconsideration of the common assumption in several prior studies that a DNN model's performance on $\mathcal{D}_f$ would deteriorate after it unlearns the provided data. However, the absence of certain data in our decision algorithm does not necessarily result in less accurate predictions.

**Remark.** *Recall that we include $ECO_i$ for comparison to validate the reasonableness of our acceleration methodology introduced in Section 3.1.1. As observed from Tables 1 and 2, the intermediate product of our methodology, $ECO_i$, trained on the compact dataset $\mathcal{C}_{\text{GD}}$, consistently outperforms NR, trained on the full dataset $\mathcal{D}_{\text{train}}$. On the other hand, Table 2 reveals that $ECO_i$ performs even better than ECO when evaluated on the retained set. This improved performance could be attributed to the fact that, unlike ECO, which utilizes a hybrid model combining both CP and GD*

*optimizers, $ECO_i$ is more finely tuned to the training data. Note that the retained set now serves as the training set for $ECO_i$.*

*On the other hand, while Tables 1 and 2 show that ECO generally achieves better accuracy than $ECO_i$, it is important to note that ECO typically requires more computational resources than $ECO_i$ due to additional calculations for $\alpha_i$ and $b$ in the CP algorithm.[7] Nevertheless, ECO's hybrid nature combines both CP and GD optimizers, facilitating unlearning specific data points and supporting continual learning. This feature ensures that ECO can adapt to evolving data and knowledge. In essence, ECO allows for the integration of new data instances without the need for retraining, effectively managing both unlearning and continuous learning adjustments.*

### 4.5.2. THE FORGETFULNESS QUALITY: MIA

Although our method qualifies as exact unlearning, ensuring complete removal of the requested data, we still evaluate forgetfulness quality through a membership inference attack (MIA) as outlined by Jia et al. (2023). To train an MIA predictor, we first sample a balanced dataset from the retained set, $\mathcal{D}_r$, and the test set, $\mathcal{D}_{\text{test}}$. Instances from $\mathcal{D}_r$ (or $\mathcal{D}_{\text{test}}$) are used to calculate losses based on the unlearned model $\theta_u$. These losses are then used as input to train a binary classifier whose task is to predict whether an instance belongs to the training set ($\mathcal{D}_r$) or the non-training set ($\mathcal{D}_{\text{test}}$).

The trained MIA predictor is subsequently employed to evaluate forgetfulness quality during its testing phase. Specifically, the MIA score is determined by applying the MIA predictor to the unlearned model ($\theta_u$) on the forgetting dataset ($\mathcal{D}_f$), defined $TN/|\mathcal{D}_f|$, where TN represents the true negatives predicted by the MIA predictor, i.e., the number of forgetting samples predicted as non-training examples.

Table 3 compares the MIA scores between $\mathcal{D}_f$ and $\mathcal{D}_{\text{test}}$ for each unlearning approach. A lower discrepancy between these two values indicates better forgetfulness quality of the model. For the NR method, there is hardly any disparity between $\mathcal{D}_f$ and $\mathcal{D}_{\text{test}}$. This pattern is also consistent for $ECO_i$ and ECO. As expected, this alignment occurs because all three methods adhere to an exact unlearning approach.

### 4.5.3. SENSITIVITY ANALYSIS ON $k_{\text{GD}}$

Referring back to Section 3.1.1, during the model preparation phase, we expedite the optimization process of the CP algorithm by reducing $\mathcal{D}_{\text{train}}$ to $\mathcal{C}_{\text{GD}}$ through the selection of the highest $k_{\text{GD}}$ loss values of the original DNN model. Figure 1 displays the accuracy scores on $\mathcal{D}_{\text{train}}$ and $\mathcal{D}_{\text{test}}$,

---

[7]The acceleration of model ECO could potentially be enhanced using techniques from Karasuyama & Takeuchi (2009), a topic we intend to explore in future research.

*Table 2.* Off-time batch unlearning

| | Acc_r ↑ | | | Acc_f ↑ | | |
|---|---|---|---|---|---|---|
| $p$ | NR | ECO_i | ECO | NR | ECO_i | ECO |
| 1 | $0.991 \pm 0.002$ | $\mathbf{0.996 \pm 0.003}$ | $\underline{0.995 \pm 0.001}$ | $0.986 \pm 0.004$ | $0.988 \pm 0.005$ | $\mathbf{0.994 \pm 0.002}$ |
| 5 | $0.990 \pm 0.002$ | $\mathbf{0.996 \pm 0.002}$ | $\underline{0.995 \pm 0.001}$ | $0.986 \pm 0.002$ | $0.988 \pm 0.003$ | $\mathbf{0.995 \pm 0.001}$ |
| 10 | $0.991 \pm 0.003$ | $\mathbf{0.996 \pm 0.003}$ | $\underline{0.995 \pm 0.002}$ | $0.985 \pm 0.002$ | $0.988 \pm 0.002$ | $\mathbf{0.994 \pm 0.002}$ |
| 20 | $0.990 \pm 0.003$ | $\mathbf{0.997 \pm 0.002}$ | $\underline{0.995 \pm 0.002}$ | $0.984 \pm 0.002$ | $0.987 \pm 0.001$ | $\mathbf{0.994 \pm 0.002}$ |
| 30 | $0.991 \pm 0.003$ | $\mathbf{0.998 \pm 0.001}$ | $\underline{0.995 \pm 0.001}$ | $0.984 \pm 0.002$ | $0.987 \pm 0.001$ | $\mathbf{0.994 \pm 0.001}$ |
| 40 | $0.992 \pm 0.002$ | $\mathbf{0.998 \pm 0.002}$ | $\underline{0.995 \pm 0.001}$ | $0.983 \pm 0.002$ | $0.986 \pm 0.001$ | $\mathbf{0.994 \pm 0.001}$ |
| 50 | $0.992 \pm 0.003$ | $\mathbf{0.997 \pm 0.002}$ | $\underline{0.996 \pm 0.002}$ | $0.982 \pm 0.002$ | $0.983 \pm 0.002$ | $\mathbf{0.994 \pm 0.002}$ |

| | Acc_test ↑ | | | Acc_all ↑ | | |
|---|---|---|---|---|---|---|
| $p$ | NR | ECO_i | ECO | NR | ECO_i | ECO |
| 1 | $0.988 \pm 0.001$ | $\mathbf{0.989 \pm 0.001}$ | $\mathbf{0.989 \pm 0.001}$ | $0.965 \pm 0.006$ | $\underline{0.974 \pm 0.007}$ | $\mathbf{0.979 \pm 0.003}$ |
| 5 | $0.987 \pm 0.001$ | $\mathbf{0.989 \pm 0.001}$ | $\mathbf{0.989 \pm 0.001}$ | $0.964 \pm 0.005$ | $\underline{0.973 \pm 0.004}$ | $\mathbf{0.979 \pm 0.002}$ |
| 10 | $0.987 \pm 0.002$ | $\mathbf{0.989 \pm 0.001}$ | $\mathbf{0.989 \pm 0.001}$ | $0.963 \pm 0.006$ | $\underline{0.974 \pm 0.005}$ | $\mathbf{0.978 \pm 0.004}$ |
| 20 | $0.986 \pm 0.001$ | $\mathbf{0.989 \pm 0.001}$ | $\mathbf{0.989 \pm 0.001}$ | $0.960 \pm 0.005$ | $\underline{0.973 \pm 0.002}$ | $\mathbf{0.978 \pm 0.004}$ |
| 30 | $0.986 \pm 0.002$ | $0.988 \pm 0.001$ | $\mathbf{0.989 \pm 0.001}$ | $0.961 \pm 0.005$ | $\underline{0.973 \pm 0.002}$ | $\mathbf{0.978 \pm 0.003}$ |
| 40 | $0.986 \pm 0.001$ | $\underline{0.987 \pm 0.001}$ | $\mathbf{0.989 \pm 0.001}$ | $0.961 \pm 0.005$ | $\underline{0.971 \pm 0.003}$ | $\mathbf{0.979 \pm 0.003}$ |
| 50 | $0.986 \pm 0.002$ | $\underline{0.986 \pm 0.001}$ | $\mathbf{0.989 \pm 0.001}$ | $0.961 \pm 0.006$ | $\underline{0.966 \pm 0.004}$ | $\mathbf{0.979 \pm 0.004}$ |

*Table 3.* The forgetfulness quality (MIA)

| | NR | | ECO_i | | ECO | |
|---|---|---|---|---|---|---|
| $p$ | $\mathcal{D}_f$ | $\mathcal{D}_{test}$ | $\mathcal{D}_f$ | $\mathcal{D}_{test}$ | $\mathcal{D}_f$ | $\mathcal{D}_{test}$ |
| 1 | $0.06 \pm 0.01$ | $0.07 \pm 0.0$ | $0.15 \pm 0.25$ | $0.16 \pm 0.24$ | $0.03 \pm 0.01$ | $0.04 \pm 0.01$ |
| 5 | $0.14 \pm 0.26$ | $0.15 \pm 0.26$ | $0.07 \pm 0.02$ | $0.08 \pm 0.02$ | $0.03 \pm 0.01$ | $0.04 \pm 0.01$ |
| 10 | $0.15 \pm 0.26$ | $0.15 \pm 0.26$ | $0.07 \pm 0.02$ | $0.08 \pm 0.01$ | $0.04 \pm 0.01$ | $0.05 \pm 0.01$ |
| 20 | $0.32 \pm 0.39$ | $0.32 \pm 0.39$ | $0.07 \pm 0.02$ | $0.07 \pm 0.01$ | $0.04 \pm 0.01$ | $0.04 \pm 0.01$ |
| 30 | $0.15 \pm 0.26$ | $0.15 \pm 0.26$ | $0.06 \pm 0.01$ | $0.07 \pm 0.01$ | $0.04 \pm 0.01$ | $0.05 \pm 0.01$ |
| 40 | $0.15 \pm 0.26$ | $0.15 \pm 0.26$ | $0.06 \pm 0.02$ | $0.07 \pm 0.01$ | $0.04 \pm 0.01$ | $0.04 \pm 0.01$ |
| 50 | $0.24 \pm 0.34$ | $0.24 \pm 0.35$ | $0.08 \pm 0.02$ | $0.08 \pm 0.02$ | $0.04 \pm 0.01$ | $0.05 \pm 0.01$ |

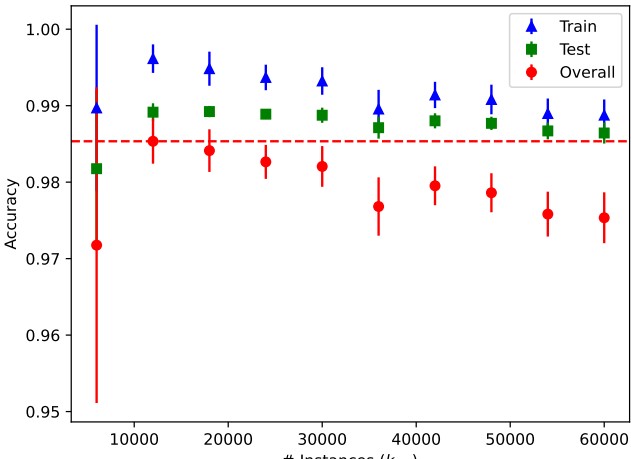

*Figure 1.* Sensitivity analysis on $k_{GD}$

along with their product, denoted as the overall accuracy, for $k_{GD} = p\% \times |\mathcal{D}_{train}|$, where $p$ ranges from 10 to 100. The mean accuracies from ten trials are represented by different symbols (△ for training data, □ for testing data, and ◯ for overall), with error bars indicating the standard deviation.

We select $k_{GD} = 12000$ as it yielded the highest overall accuracy score. Interestingly, using the full training dataset,

where $k_{GD} = 60000$, leads to poor performance.[8]

## 5. Conclusion

Our proposed method, ECO, represents a significant advancement in exact unlearning within DNNs, preserving model utility without compromising performance. Integrating the CP algorithm into machine unlearning for deep neural networks (DNNs), ECO combines the robust data-driven capabilities of neural networks alongside the elegant characteristics of the CP algorithm. While not exhaustive, the positive outcomes from ECO suggest its effectiveness. We also introduce ECO_i, an intermediate version of ECO that is easier to implement and outperforms the state-of-the-art method SISA (Bourtoule et al., 2021), which has previously been reported to sacrifice performance compared to naive retraining. Therefore, ECO_i sets as a new benchmark in exact unlearning, potentially reducing the need for full retraining—a vital consideration in the context of climate change. This study also encourages further exploration into dual-domain strategies, analogous to the primal and dual formulations in SVMs, in machine unlearning within DNNs.

---

[8]This case mirrors the scenario without the acceleration method mentioned in Section 3.1.1.

## Acknowledgments

This work was supported in part by the National Science and Technology Council of Taiwan under Grant NSTC-112-2221-E-002-204-(112C6220), the Administration for Digital Industries, moda, R.O.C. (Taiwan) under Grant NTU112HT911011, the Asian Office of Aerospace Research & Development (AOARD) under Grant NTU-112HT911020, and the financial supports from the Featured Area Research Center Program within the framework of the Higher Education Sprout Project by the Ministry of Education (113L900901/113L900902/113L900903).

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
