# OpenReview forum: "ECO: Efficient Computational Optimization for Exact Machine Unlearning in Deep Neural Networks"
_ICML.cc/2024/Workshop/WANT — WANT@ICML 2024 Poster_

### Official Review · Reviewer_Hv3o · 2024-06-12
**CP algorithm with DNNs for exact machine unlearning**

**Confidence:** 2

**Summary:**

This paper integrated CP algorithm with DNNs for exact machine unlearning.
It shows a large improvement in unlearning efficency but I am not sure if this paper matches the topic of the workshop.

**Strengths:**

Innovative method on improving the machine unlearning method. Well-explained algorithm and clear analysis.

**Weaknesses:**

Lack of model details of DNN models used for the experiments.

Only compared the performance with naive retaining. Are there any other approaches to compare with?

Only the difference between MIA scores on the test set and the forget set is analyzed. What can we learn from the absolute values that reflect the percentage of samples predicted as non-training examples?

---

### Official Review · Reviewer_1Q4v · 2024-06-13
**Good insight and approach statement with somehow inadequate experimental results**

**Confidence:** 3

**Summary:**

The paper presents an approach, ECO, to unlearn exact data points from the training set while mitigating model utility loss. To validate their approach's effectiveness, they evaluate their approach and compare it with other existing approaches.

**Strengths:**

1.	The paper describes their approach clearly.
2.	The paper shows the evolutions under various scenarios, making it more convincing.

**Weaknesses:**

1.	The choice of Datasets for evaluation: machine unlearning aims to avoid the cost of training from scratch when the training cost is too high on large models and datasets. However, the dataset and model used in the paper are tiny, leading to unimpressive results during comparison.

**Suggestions:**

1.	More datasets to show the disparity between approaches.

---

### Meta-Review · Area_Chair_koCt · 2024-06-17

**Recommendation:** Accept (Poster)
**Confidence:** 3

**Metareview:**

The manuscript extends the CP algorithm to the field of machine unlearning and achieves efficient and effective unlearning via a single model. Though all reviewers acknowledge the paper's writing quality and methodology novelty, they criticize the empirical evaluation issues due to the limited/small-scale neural architecture, evaluation datasets, and baselines. The AC would suggest authors to further justify the effectiveness of the proposed method via extensive evaluations.

---

### Decision · Program_Chairs · 2024-06-17

**Decision:**

Accept (Poster)

**Comment:**

We thank the authors for their time and contribution to WANT and we are pleased to share that after the reviewing process the paper has been accepted. Congratulations! We encourage the authors to consider reviewers' feedback for the improvement of the camera-ready version. We hope to see you in person at the workshop and brainstorm on efficient training research together!